# Investigating the Aftershock of a Disaster: A Study of Health Service Utilization and Mental Health Symptoms in Post-Earthquake Nepal

**DOI:** 10.3390/ijerph16081369

**Published:** 2019-04-16

**Authors:** Tara Powell, Shang-Ju Li, Yuan Hsiao, Chloe Ettari, Anish Bhandari, Anne Peterson, Niva Shakya

**Affiliations:** 1School of Social Work, University of Illinois Urbana—Champaign, 1010 W. Nevada St., Urbana, IL 61801, USA; 2Americares, 88 Hamilton Avenue, Stamford, CT 06902, USA; sjli@americares.org (S.-J.L.); cettari@elm-project.org (C.E.); apeterson@americares.org (A.P.); 3Department of Sociology, University of Washington, Seattle, WA 98195-3340, USA; yahsiao@uw.edu; 4Social Science Baha, 345 Ramchandra Marg, Battisputali, Kathmandu, Nepal; anish.bhandari@ymail.com; 5Americares Nepal, Dhobighat 4, Lalitpur, State 3, Nepal; nshakya@americares.org

**Keywords:** disaster, mental health, access to care, healthcare utilization

## Abstract

*Background*: In 2015, a 7.8 magnitude earthquake struck Nepal, causing unprecedented damage and loss in the mountain and hill regions of central Nepal. The aim of this study was to investigate the association between healthcare access and utilization, and post-disaster mental health symptoms. *Methods*: A cross-sectional study conducted with 750 disaster-affected individuals in six districts in central Nepal 15 months post-earthquake. Anxiety and depression were measured through the Depression, Anxiety and Stress Scale (DASS-21). Healthcare utilization questions examined types of healthcare in the communities, utilization, and approachability of care providers. Univariate analyses, ANOVAs and Tobit regression were used. *Results*: Depression and anxiety symptoms were significantly higher for females and individuals between 40–50 years old. Those who utilized a district hospital had the lowest anxiety and depression scores. Participants who indicated medical shops were the most important source of health-related information had more anxiety and depression than those who used other services. Higher quality of healthcare was significantly associated with fewer anxiety and depressive symptoms. *Conclusions*: Mental health symptoms can last long after a disaster occurs. Access to quality mental health care in the primary health care settings is critical to help individuals and communities recover immediately and during the long-term recovery.

## 1. Background

On 25 April 2015, a 7.8 magnitude earthquake struck Nepal, causing unprecedented damage and loss in the mountain and hill regions in the center of the country [1]. The government-commissioned Nepal Earthquake Post-Disaster Needs Assessment documented significant losses and damages to private property, farmland, and livestock. Damages from the earthquake are estimated at USD 5.15 billion, losses at USD 1.9 billion, and recovery needs at USD 6.6 billion, roughly a third of the country’s economy [2]. Further, it is estimated that approximately 1 million people were forced into poverty as a result of the earthquake. There were 8702 earthquake-related deaths and injuries to 22,303 people. Property damage included the collapse of 498,852 houses, damage to an additional 256,697, and the destruction of up to 90% of health facilities in the highest affected areas [2,3].

The mental health impact of a disaster such as the Nepal earthquake can have far-reaching consequences. Immediately after a traumatic event, individuals may experience an array of reactions such as anxiety, depression, shock, dissociation, and agitation [4]. The long-term mental health impact can range from minimal symptoms to severe persistent psychological distress such as posttraumatic stress disorder, depression, and anxiety [4,5]. Factors such as recent and past life stressors, gender, age, level of trauma exposure, education, income change, social support, and frequency of chronic disease predict the development of mental health symptoms following trauma [6]. Co-occurring symptoms such as depression and substance abuse are also high after a disaster, and research has illustrated a causal association between mental and physical health problems, particularly among vulnerable populations [7,8]. Moreover, scholars have noted that disaster recovery can be impeded when a large portion of the community is experiencing the psychological impact of a disaster [9]. Given the impact of mental health among disaster-affected individuals, humanitarian and disaster response organizations have prioritized mental health and psychosocial support services (MHPSS) in their disaster response and recovery efforts.

### 1.1. Mental Health in Nepal

A variety of barriers impede mental health care delivery in low- and middle-income countries (LMICs) such as Nepal. Healthcare delivery in developing countries generally suffers from a severe shortage of human resources [10], and there is a dearth of mental health professionals and policies for provision of MHPSS in LMICs [11]. Recent estimates from the World Health Organization [12] indicate that low-income countries have approximately 0.16 psychiatric nurses and 0.05 psychiatrists per 100,000 people. Similar to other LMICs, Nepal, experiences a severe shortage of mental health services with most mental health services only available in large metropolitan areas such as Kathmandu. Moreover, prevalence rates of mental health professionals indicate and there are approximately 0.22 psychiatrists and 0.06 psychologists per 100,000 people [13]. Specialized mental health care in Nepal can only be received in tertiary care settings. Therefore, psychiatrists, psychologists, mental health nurses, social workers or other trained specialists are not available at the district level and people often have to travel to large cities such as Kathmandu to receive mental health treatment [13].

Misconceptions about mental health often leads to discrimination and prejudice in much of East and Southeast Asia [14]. Stigma attached to both psychiatric institutions and individuals affected by mental illness can impede service utilization [14]. In Nepal, the label of “mental health problem” carries a heavy social stigma [15]. For example, in Nepal, those who suffer from mental illness acquire a social label of a person with a “psychological problem”. Therefore, it is common to utilize a traditional healer when symptoms first present and call on a psychiatrist or psychologist as the last resort [15]. Traditional healers are generally more accessible in rural communities given the lack of availability of doctors or district hospitals [16]. Traditional healers generally view mental and physical health complaints for the person through the loss of the soul. They focus on the realignment of the relation of the human and spirit world to treat psychological distress including nightmares, fear, fatigue or depression [15]. 

Private pharmacies and medical shops are also often used in Nepal to treat mental health related issues. Medical shops for example, provide prescription and allopathic medicines such as herbal remedies [17]. Private pharmacies typically provide medicines and health information but are not generally trained in mental health [18]. In rural areas of Nepal there is not much difference in the private pharmacies and medical shops. They are generally run by the paramedics or auxiliary health workers who have studied medicine for 1 to 3 years. While these workers have professional knowledge and certification to prescribe medicines, they are generally not trained in mental health [13]. 

### 1.2. Mental Health Post-Disaster Nepal

After the earthquake in Nepal in 2015, there was a coordinated effort to address the mental health of survivors. A psychosocial support sub-cluster led by the ministry of health and the World Health Organization was established to assess mental health needs and coordinate activities. This sub-cluster was part of the larger cluster approach designed to bring together local and international governmental and non-governmental organizations to address the health and protection needs of earthquake survivors [19]. Types of mental health services that were made available included mobile health clinics, training of health workers to provide basic mental health support in primary care clinics, school and community-based interventions, public awareness initiatives, and a counseling hotline [19]. Mobile health clinics, for example, offered temporary counseling and outreach to remote earthquake affected regions. School and community-based interventions focused on empowering parents to become involved in school activities, and training teachers in positive discipline approaches to create a respectful and safe learning environment. Awareness initiatives involved raising public awareness on mental health issues and reducing stigma [19].

Specific trainings such as the World Health Organization’s Mental Health Gap Action humanitarian intervention guide (mhGAP-HIG) were implemented to equip primary care nurses and doctors with the knowledge and skills to treat those who are experiencing mental health symptoms [20]. mhGAP-HIG includes training modules focused on identification and treatment of mental, neurological, and substance use disorders. The underlying premise for mhGAP-HIG is that many people with mental disorders, especially in LMICs, do not have access to or seek targeted mental health care, and are more likely to obtain care from primary health providers [14,21]. Through education, mhGAP-HIG aimed to build the capacity of local healthcare providers beyond the disaster response and recovery period.

While strides have been taken to address the mental health needs of disaster affected communities, a dearth of research has examined how access, perceived quality, approachability, and utilization of different healthcare settings may be associated with mental health outcomes. To address this gap in research, our study examines the relationship between mental health symptoms, perceived quality, and utilization of primary health care in disaster-affected communities in Nepal. Our primary research aim was to examine the correlation between depression and anxiety symptoms with perceived quality, approachability, and utilization of healthcare services. Additionally, we explored the prevalence of depression and anxiety symptoms 15 months after the earthquake struck Nepal based on demographic characteristics including gender and socio-economic status.

## 2. Methods

This study presents the first wave of data from a larger longitudinal research study examining the impact of a mental health and psychosocial intervention in post-earthquake Nepal. From 67 village development communities (VDC) in six earthquake affected districts in Nepal, 15 were selected based on the level of impact. Sampling of VDCs was completed using the Probability Proportionate to Size (PPS) method as well as United Nations Office for the Coordination of Humanitarian Affairs (OCHA) [22] ratings of the earthquake impact in each VDC. In each VDC, 50 households were selected using systematic random sampling. This was done by selecting the first household closest to the city center and then every alternate household in the 0.5 km radius until 50 individuals in each VDC were included. The study was completed through collaboration with AmerCares, a global health organization and Social Science Baha (SSB), a Nepali research institute [23]. Three trained researchers from Social Science Baha, a Nepali research organization, collected data in August of 2016, 15 months after the earthquake. All surveys were translated to Nepali by SSB and surveys were read by the trained researchers to the respondents to ensure comprehension. 

All study participants completed consent forms to take part in the study and the National Health Research Committee (NHRC) of Nepal approved the study activities. The NHRC is a national body responsible for providing scientific study and quality health research in the country with the highest level of ethical standards. Its primary responsibility is to promote and coordinate health research with the objective of improving the health status of the people of Nepal. It also acts as a national regulatory body for maintaining technical and ethical standards of health research within the country. Thus, NHRC conducts an ethical review of all research protocols that researchers, national authorities, non-government agencies, and academic institutions intend to implement in the country.

### 2.1. Measures

The Depression Anxiety Stress Scale (DASS) 21 is an abbreviated version of the DASS 45 questionnaire, which measures a range of symptoms associated with depression and anxiety. The DASS elicits answers related to the presence of a symptom over the previous week. Each item is scored from 0 (did not apply to me at all over the last week) to 3 (applied to me very much or most of the time over the past week; 24). The range of scores for depression symptoms are 0–4 normal, 5–6 mild, 7–10 moderate, 11–13 severe and 14+ extremely severe. Scores for anxiety are 0–3 normal, 4–5 mild, 6–7 moderate, 8–9 severe, and 10+ extremely severe [24,25]. The DASS has illustrated strong reliability in previous samples, ranging from 0.72 to 0.91 [24,26]. To ensure relevance of the scale to the Nepali context, Social Science Baha researchers translated the items into Nepali to suit language and cultural codes. Further, the Cronbach Alpha test was conducted to test the reliability of the adapted scales. This revealed high Cronbach’s Alpha for both the depression (0.92) and anxiety (0.91) symptom subscales of the adapted version of the DASS 21.

Demographic items were also collected, including gender, age, income, literacy, and impact of the earthquake (e.g., loss of family member, injury, loss of home). Because most of the study sample was in highly rural settings where many individuals may be literate but did not attend school, educational attainment included the following response options for level of schooling finished: (a) illiterate; (b) literate but no formal school; (c) primary; (d) secondary; and (e) bachelor’s plus. This response set follows Gan et al. [27] and Tinkers’ [28] recommendations for measuring educational attainment in developing countries. Moreover, the questionnaire was designed as such because of the importance of distinguishing literacy among people who have not attended school.

The researchers developed healthcare access questions to identify types of available health care, quality of services, and utilization. Two of the questions included categorical responses: (1) “What type of healthcare do you make use of most” (response set: (a) health post, (b) traditional healer, (c) district hospital, (d) FCHVs, (e) private pharmacy, (f) private clinic/hospital, (g) other); (2) “Who is the most important source of information about health issues” (response set: (a) doctor, (b) health assistant, (c) health worker, (d) nurse midwife, (e) FCHV, (f) traditional healer, (g) other). Likert scale questions were also asked inquiring: (1) “What is the quality of care in the healthcare delivery system in your community” (response set: 1 = very bad to 5 = excellent); and (4) “How approachable are health service providers” (response set: 1 = not approachable to 4 = very approachable. One dichotomous variable was also included asking: “Have you talked to anyone about your current emotional status” (response set: 1 = yes, 2 = no). 

### 2.2. Study Sample

The sample included 750 community members from 15 VDCs in the three districts. Table 1 shows the distribution of socio-demographic variables. The sample included 532 (70.9%) females and 218 (29.1%) males. Most of the sample population were less than 30 years old (29.1%). Regarding the education level, 245 (32.7%) interviewees reported that they were illiterate, 158 (21.1%) reported that they were literate but no formal school, and 19 (2.5%) finished university or higher degrees. For socioeconomic status, <20,000 Nepali Rupees (NPR) (approximately 195 US dollars) was considered the poverty rate for household income. 147 (19.6%) respondents chose not to answer this question. Almost three quarters, 531 (70.8%) respondents reported they earned less than 20,000 NPR annually. Only 72 (9.6%) respondents reported that they earned more than 20,000 NPR. In terms of the impact of earthquake, the majority reported house or property damage: 728 (97.1%) of interviewees reported their house was destroyed and the remainder reported more minor property damage. A small minority, 48 (6.4%), reported that at least one family member died, and 111 (14.8%) reported that at least one of their family members had been injured during the earthquake. See Table 1 for demographic characteristics.

### 2.3. Analyses

The data were first cleaned and a series of frequency tables/diagrams developed, which provided the basis for further analysis of the available data. A univariate analysis was performed for basic socio-demographic and health service utilization variables. We calculated scores for depression and anxiety by summing up the items, and tested the reliability using Cronbach’s Alpha. Finally, we assessed the average value of depression and anxiety symptoms across a series of socio-demographic variables.

ANOVA was used for the comparison of categorical independent variables. Since not all variables meet the equality of variances assumption, we used the Welch correction [29] and the Dunnett T3 test [30] for post-hoc comparison, which is a conservative test (i.e., has higher probability of false acceptance of the null hypothesis). Post-hoc comparison was not conducted for groups with unstable standard errors (e.g., group size < 2). We conducted separate analyses for the whole sample, for males and females, and for those above and below the poverty line. The analyses therefore allowed for examination of associations within these specific populations. The statistical software SPSS 19 (IBM, Armonk, NY, USA) was used for estimation. Tobit regression was used for ordinal independent variables, since both the scales of anxiety and depression symptoms have absolute minimums. The statistical software *R* (*R* version 3.5.1 Vienna, Austria)*,* with the *AER* package (*AER* package version 1.2–5, Vienna, Austria) was used for estimation [31,32].

## 3. Results

The average depression symptom score of the entire sample was *M* = 9.64, *SD* = 3.61 and the average anxiety score was *M* = 10.79, *SD* = 3.93 (See Table 2). Anxiety scores were significantly higher (*p* = 0.05) in female participants (*M* = 10.97, *SD* = 3.87) than in male participants (*M* = 10.34, *SD* = 4.04). Depression symptom scores were also significantly (*p* < 0.01) higher in females (*M* = 9.88, *SD =* 3.64) than in males (*M =* 9.07, *SD* = 3.48). Among the age groups, anxiety and depression symptoms were highest for those 40–50 years old (anxiety symptoms *M* = 11.89, *SD* = 4.97; depression symptoms *M* = 10.30, *SD* = 4.06). The differences in depression and anxiety symptoms did not correlate significantly with income level. Moreover, we found that 633 (58.2%) of respondents approached someone to discuss their current mental health status. 

We first examined differences in health seeking behavior by gender and income on depression and anxiety symptom scores through ANOVA comparisons (see Table 3). The results revealed significant differences in the type of healthcare individuals make the most use of between groups for anxiety F (7, 38.87) = 3.52, *p* = 0.005, but non-significant for depression symptoms F (7, 39.08) = 2.16, *p* = 0.060. We also found that participants’ source of information for health issues were significantly correlated with differences in anxiety F (7, 95.21) = 2.20, *p* = 0.041 and depression symptoms F (7, 94.86) = 3.34, *p* = 0.003 (see Table 3).

Chi Square tests then examined differences between age, gender, and income on type of healthcare facility that was used. We found that significant differences existed between income *x*^2^ (7, 750) = 27.13, *p* < 0.001 and age *x*^2^ (7, 750), *p* < 0.01 on type of healthcare facility. Those who had lower income were more likely to use the health post (34.9%) and traditional healers (8.8%) compared to those who had a higher income (≥20,000 NPR). Those who were over 60 years old were more likely to use the health post. Those who were above 30 years old were more likely to use the district hospital compared to those older than 30, and those who were between 40–50 and above 60 were less likely to use private clinics/hospitals than those in other age ranges (see Table 4).

We then examined within group differences of gender and income to explore the correlation between type of healthcare and source of healthcare on anxiety and depression symptoms through the Welch correction, and the Dunnett T3 test for post-hoc comparison. People who chose a district hospital as their major medical source reported significantly lower anxiety symptom scores than people who chose a traditional healer (M = 1.5, SE = 0.34), private pharmacy (M = 1.61, SE = 0.42), or private clinics/hospital (M = 1.40, SE = 0.42).

Depression symptom scores displayed a similar pattern. People who chose a district hospital as their major medical source reported significant lower depression symptom scores than those who utilized traditional healers (M = 1.12, SE = 0.37) and private pharmacies (M = 1.60, SE = 0.46). We also tested gender and income to examine differences in groups. Women (M = 1.81, SE = 0.47) and those with higher incomes (M = 1.26, SE = 0.42) who used a traditional healer had significantly higher depression symptoms than those who used the district hospital. We also found those with higher income had significantly (M = 2.03, SE = 0.55) higher depression symptom scores when using a private pharmacy (see Table 5). Those who chose medical shops (i.e., shops selling medicines and herbal remedies) as the most important source of health-related information had significantly higher anxiety (M = 2.05, SE = 0.60) and depression symptom (M = 2.22, SE = 0.53) scores than those who chose Female Community Health Volunteers (FCHVs). We also found that individuals below the poverty line (NPR < 20,000) had significantly higher depression symptoms when they indicated doctors (M = 1.44, SE = 0.44) and health workers (M = 1.31, SE = 0.46) were the most important source of information (see Table 5).

Tobit regression results (see Table 6) indicated those who perceived a higher quality of the health care system in their community had significantly lower depression (*p* < 0.001) and anxiety symptom (*p* < 0.001) scores than those who do not. Examining differences in income and gender suggest that men (*p* < 0.001) and those with lower incomes (*p* < 0.01) drive these results with respect to anxiety symptoms, as they reported significantly higher anxiety symptom scores when they perceived quality of healthcare was lower. Examining differences in income and gender showed that men (*p* < 0.001) and those with lower incomes (*p* < 0.01) reported significantly higher anxiety symptom scores when perceived quality of healthcare was lower. Lower depressive symptoms were significantly (*p* < 0.05) correlated with those who felt their healthcare providers were more approachable. The more approachable the healthcare providers was significantly (*p* < 0.05) correlated with fewer depressive symptoms in women than men, but income level did not predict any correlation. 

## 4. Discussion

To our knowledge, this study is the first to examine the relationship between perceived quality, approachability, and healthcare utilization on mental health outcomes in a disaster-exposed sample in a low and middle-income country (LMIC). Although it was 15 months after the earthquake, depression and anxiety symptoms remained high, ranging from moderate to extremely severe among disaster affected individuals. Our results indicated anxiety and depression symptoms were correlated with type of healthcare facility, perceived approachability of healthcare providers, and perceived quality of healthcare. These findings support the literature suggesting that the quality and access of MHPSS services may have an impact on mental health outcomes [33]. Following, we will discuss our main findings from the paper including utilization of healthcare and quality of care. Implications for future research will also be discussed.

### 4.1. Utilization

#### 4.1.1. District hospitals 

Those with higher incomes used the district hospitals more than those who earned below 20,000 NPR. This may be due to cost as those below poverty level may not be able to afford or access doctors through district hospitals. Therefore, impoverished individuals and families may be more likely to use other forms of healthcare such as traditional healers than doctors or health workers regarding their mental health symptoms. We also found those who use the district hospital had significantly lower depression and anxiety symptoms. This may be attributed to access to medication at the district hospital or the presence of health professionals with training in supporting individuals who are experiencing depression and anxiety symptoms. These findings are particularly applicable because of a trend among humanitarian organizations of training community-based primary health care workers in mhGAP-HIG. Training doctors and other medical professionals in LMICs to identify and treat those with mental health issues may be particularly vital in Nepal, as many Nepalis do not have access to district level health centers due to distance, financial constraints, and difficult terrain. Assessing and understanding the capacity of those at the district level could inform training of health workers in MHPSS at the local level such as in VDCs and municipalities. 

#### 4.1.2. Traditional healers 

Those with lower incomes were more likely to use traditional healers than those who earned above 20,000 NPR. This may be due to access or cost. For example, those who lived in higher poverty areas were also in remote regions where they may not have had access to district hospitals. Additionally, they may not have been able to afford private pharmacies or private hospitals. Individuals who used traditional healers as their primary source of healthcare also had significantly higher anxiety and depression symptom scores than those who went to the district hospital. Nepalis widely use traditional healers, especially in remote areas of the country, to heal mental and physical health difficulties. But they often do not have training in MHPSS [15]. Considering traditional healers are often the frontline of care for physical and mental health symptoms in remote areas where communities might not have access to other services, training them in basic MHPSS or when to refer those with more serious conditions may be beneficial. As previous studies have noted, collaboration with traditional healers may also break down barriers to mental health treatment and delivery [34]. 

#### 4.1.3. Medical shops 

Reliance on medical shops as a primary source of health information also correlated with depression and anxiety symptoms. Medical shops are widely used in Nepal for both prescription and allopathic medicines such as herbal and ayurvedic remedies [17]. Those who run the medical shops often do not have professional training yet provide prescriptions and medical advice to consumers. Considering the common utilization of medical shops, efforts have been made to provide professional training to these individuals for physical health ailments [35]. Mental health training for medical shop employees to increase their capacity in providing MHPSS information to patients could also benefit Nepalis. 

### 4.2. Quality of Care

#### 4.2.1. Approachability

Results indicate the approachability of the care provider was significantly correlated with lower depression and anxiety symptom scores. As organizations and governments are scaling up and strengthening mental health services, it is critical to take these findings into consideration. While it is important to provide services that have a scientific evidence base, it is also essential for care providers to be approachable and build a level of trust with the individual and community. Moreover, as previous studies have shown, lack of trust or approachability can impede service utilization [36]. 

#### 4.2.2. Perceived quality

Perceived higher quality of healthcare was significantly associated with lower depression and anxiety symptom scores. This correlation supports previous research on the intersection of quality of care and patients’ well-being [37,38]. Given this association it may be important to consider integrating systems with a focus on quality improvement to increase the capacity of mental health care in LMICs. As Patel and colleagues [38] note, the building blocks to establish effective mental health systems include a complex interplay of engagement, education, evidence-based practices, pharmacological treatment, and continuous quality improvement. 

### 4.3. Limitations

One limitation of this study was that it was conducted 15 months after the earthquake. Some of the health systems participants had access to pre-disaster may not have been accessible post-earthquake. Therefore, we did not have data on access or barriers to health systems in the regions where the study took place, which likely impacted they type of healthcare participants used.

The cross-sectional analysis was a second limitation to the study. While the outcomes established correlations, they cannot assume causality. The data used for this manuscript was embedded in a larger longitudinal study, however we did not have data prior to 15 months after the earthquake. Therefore, depression and anxiety scores may have been much higher directly following the earthquake. Future studies might gather data directly post-disaster and examine access across time to determine change in symptomology based on health care facility. A third limitation was the lack of knowledge about mental health education in each facility. After the earthquake, humanitarian organizations delivered the mhGAP program in addition to other MHPSS trainings. It is likely the training occurred at the district hospital level. Some individuals at the district hospitals had knowledge on treating those with mental health issues; however, without the data we do not know the level of training at each facility that is utilized by our study participants. Finally, while we found higher depression and anxiety in individuals who used traditional healers and medical shops, the results should be treated with caution. Confounding factors such as location and access may have influenced our findings considering many individuals who use these forms of care live in rural settings where other forms of healthcare may not be available.

The limitations of the current study are common to post-disaster research. As scholars have noted, the inability to pre-plan for a disaster, the upheaval and constantly changing conditions of the environment, and the difficulties of coordinating an efficient and qualified research team all complicate post-disaster research [39].

### 4.4. Implications

In the past ten years there has been a movement to provide quality mental health interventions in LMICs. Our study lends to the research examining the connection between quality, type of healthcare used, and approachability of providers on mental health symptoms. Given our findings, implications and potential areas of future research should be considered. 

Many Nepali’s live in remote areas where access to services may be limited and there may be lack of awareness on mental health issues. Future research might examine strategies and best practices to reach this group, as many do not use services due to stigmatization of mental health services [33].

Additionally, considering many of our participants indicated they use non-specialized resources (e.g., traditional healers, district hospitals, medical shops) for mental health related issues, training is needed to enable these providers to identify, provide general support, and be able to refer those with more severe psychosocial needs to appropriate care. As previous studies have suggested, there is need for a collaborative approach to care, particularly in settings such as Nepal where the mental health workforce capacity is low, and the mountainous terrain makes healthcare access difficult [40]. 

Considering the treatment gap for those who live in poverty, community-based models which integrate local social service and healthcare interventions to reach those is low resource environments might greatly enhance the quality of care. As Raja et al. [41] note, this could include incorporating local resources, and involve affected individuals, families, and communities to provide a sustainable model of care for individuals with mental health needs. Future research should also examine the impact more generalized psychosocial support rather than focusing primarily on treating those with severe mental illnesses. As mental health system capacity continues to evolve in LMICs it is critical to ensure quality of care for individuals at all levels. Training might include psychological first aid or other informational sessions designed for non-mental health professionals. These sessions could include information on common reactions to traumatic stressors, psychosomatic symptoms related to distress, and basic helping skills. Future studies should examine strategies and practices to build mental health support capacity at the local levels with lay community health workers. Training community health workers may in turn reach those who do not have access to healthcare on the district level.

## 5. Conclusions

Considering much of Asia is vulnerable to natural disasters such as earthquakes, floods, or landslides, it is critical to understand the impact utilization of health care facilities on mental health outcomes [14]. The Grand Challenges for Global Mental Health Initiative, established by a consortium of researchers, clinicians, and advocates has identified one of the priorities in the next ten years is to improve treatment and expand access to care in LMICs [10]. This involves training lay professionals, increasing the usage of evidence-based or evidence informed interventions, and providing effective community-based care [10]. Historically, lack of training in MHPSS among primary health-care staff and shortage of mental health providers to supervise lay community workers have been identified as key concerns to advancing mental healthcare [33]. Our study suggests type and perceived quality of health care may play a role in mental health symptoms during the long-term recovery. Presently, mental health services can be challenging in a post-disaster context, but with continued research, humanitarian organizations and governments can enhance service delivery, develop policy initiatives, and disseminate evidence-based practices to improve mental health symptoms for all-disaster affected communities.

## Figures and Tables

**Table 1 ijerph-16-01369-t001:** Demographic characteristics.

Characteristics	N	%
Gender		
Male	218	(29.07)
Female	532	(70.93)
Age		
<30 Years	218	(29.07)
30–40 Years	147	(19.6)
40–50 Years	127	(16.93)
50–60 Years	137	(18.27)
>60 Years	121	(16.13)
Literacy		
Illiterate	245	(32.67)
Literate but no formal school	158	(21.07)
Primary	120	(16.00)
Higher secondary	208	(27.73)
Bachelor’s plus	19	(2.53)
Income		
Below poverty (NPR <20,000)	531	(70.8)
Above poverty (NPR >20,000)	72	(9.6)
Impact of Earthquake		
Death of family member(s)	48	(6.40)
Injury to family member(s)	111	(14.80)
Loss of employment	7	(0.93)
Loss of sources of livelihood	13	(1.73)
House destroyed	728	(97.07)
Damage of other property	199	(26.53)
Loss of agricultural land	21	(2.80)
Loss of livestock	214	(28.53)

**Table 2 ijerph-16-01369-t002:** Depression and anxiety symptom score.

Demographics	N	Depression Score	SD	%	N	Anxiety Score	SD	%
Overall	750	9.64	3.61	100%	750	10.79	3.93	100%
Gender								
Male	218	9.07	3.48	29%	218	10.34	4.04	29%
Female	532	9.88	3.64	71%	532	10.97	3.87	71%
Age								
<30 Years	186	9.04	3.38	25%	186	10.03	3.51	25%
30–40 Years	147	9.36	3.34	20%	147	10.39	3.70	20%
40–50 Years	129	10.30	4.06	17%	129	11.89	4.97	17%
50–60 Years	147	9.99	4.02	20%	147	11.14	3.87	20%
>60 Years	141	9.77	3.19	19%	141	10.84	3.41	19%
Literacy								
Illiterate	245	10.33	3.96	33%	245	11.54	4.18	33%
Literate but no formal school	158	9.78	3.47	21%	158	11.16	4.11	21%
Primary	216	9.30	3.52	29%	216	10.40	3.94	29%
Secondary	112	8.79	2.87	15%	112	9.58	2.57	15%
Bachelor’s plus	19	8.47	3.56	3%	19	9.58	3.31	3%
Income								
<20,000 NPR	531	9.68	3.75	71%	531	10.74	3.94	71%
>20,000 NPR	72	9.50	3.64	10%	72	10.33	3.87	10%

**Table 3 ijerph-16-01369-t003:** ANOVA Results: between group differences on gender and income.

Health Seeking Behavior	*Anxiety*		*Depression*	
	*df*	F	*df*	F
**Which type of healthcare do you make use of most?**				
Total	7, 38.87	3.52 **	7, 39.08	2.16
Male	7, 13.64	1.50	7, 15.12	5.92 **
Female	7, 12.65	4.13 *	7, 12.70	2.76
Income < 20,000 NPR	5, 168.39	2.90 *	5, 170.31	3.49 **
Income ≥ 20,000 NPR	5, 27.23	3.11 *	5, 28.42	7.57 ***
**Who is the most important source of information about health issues?**				
Total	7, 95.21	2.20 *	7, 94.86	3.34 **
Male	7, 57.33	1.32	7, 57.08	1.70
Female	7, 27.45	1.09	7, 27.94	2.12
Income < 20,000 NPR	4, 94.28	4.28 **	4, 97.09	2.09
Income ≥ 20,000 NPR	4, 23.83	2.88 *	4, 27.32	5.69 **

* *p* < 0.05; ** *p* < 0.01, *** *p* < 0.001.

**Table 4 ijerph-16-01369-t004:** Demographic differences between healthcare utilization.

Characteristics	Health Post		Traditional Healers		District Hospital		FCHVs		Private Pharmacy		Private Clinics/Hospital		Others		I Don’t Use Any Type of Health Facility		*X* ^2^
	N	%	N	%	N	%	N	%	N	%	N	%	N	%	N	%	
Female	176	33.0%	42	7.8%	53	9.9%	2	0.00%	84	15.7%	115	21.6%	57	10.7%	3	0.1%	5.25*_ns_*
Male	77	35.3%	18	8.2%	27	12.3%	3	1.3%	28	12.8%	41	18.8%	22	10.1%	2	0%	
<30 Years	64	29.3%	14	6.4%	16	7.3%	1	0.1%	28	12.8%	50	22.9%	13	5.9%	0	0%	55.38 **
30–40 Years	45	30.6%	16	10.8%	20	13.6%	1	0.01%	22	14.9%	30	20.4%	13	8.8%	0	0%	
40–50 Years	45	35.4%	14	11.0%	15	11.8%	1	0.01%	19	14.9%	20	15.7%	15	11.8%	0	0%	
50–60 Years	46	33.5%	10	7.2%	16	11.6%	0	0%	22	16.1%	36	26.2%	17	12.4%	0	0%	
60+ Years	53	43.8%	6	4.9%	13	10.7%	2	1.6%	21	17.3%	20	16.5%	21	17.3%	5	4.1%	
Below poverty (NPR < 20,000)	237	34.9%	60	8.8%	64	9.4%	5	1%	98	14.4%	134	19.7%	75	11.0%	5	1%	27.13 ***
Above poverty (NPR > 20,000)	16	22.2%	--	0%	16	22.2%	--	0%	14	19.4%	22	30.5%	4	5.5%	0	0%	

** *p* < 0.01, *** *p* < 0.001.

**Table 5 ijerph-16-01369-t005:** Within group comparisons by type and source of healthcare.

Type and Source of Healthcare	Anxiety					Depression				
	Total	Male	Female	Income < 20,00 NPR	Income ≥ 20,00 NPR	Total	Male	Female	Income < 20,000 NPR	Income ≥ 20,00 NPR
**Which type of healthcare do you make use of most?**										
District Hospital (Ref)	(M = 9.44)	(M = 9.96)	(M = 8.41)	(M = 9.55)	(M = 9.16)	(M = 8.69)	(M = 9.28)	(M = 7.52)	(M = 8.55)	(M = 9.37)
	MD (SE)	MD (SE)	MD (SE)	MD (SE)	MD (SE)	MD (SE)	MD (SE)	MD (SE)	MD (SE)	MD (SE)
Traditional Healer	1.5 (0.34) ***	0.964 (0.427)	2.54 (0.589) *	2.07 (1.06)	1.13 (0.62)	1.12 (0.37) *	0.85 (0.48)	1.81 (0.47) *	1.13 (1.12)	1.26 (0.42) *
Health post	1.45 (.64)	1.30 (0.873)	1.59 (0.661)	0.44 (0.82)	0.80 (0.80)	0.73 (0.54)	0.48 (0.70)	1.09 (0.71)	0.03 (1.87)	0.66 (0.59)
FCHVs	2.96 (2.12)	5.54 (3.52)	1.93 (2.34)	NA	NA	0.71 (1.39)	0.22 (1.56)	1.82 (2.35)	NA	NA
Private Pharmacy	1.61(0.45) **	1.19 (0.514)	2.31 (0.96)	1.09 (0.72)	1.80 (0.70)	1.60 (0.46) *	1.00 (0.55)	2.77 (0.87)	−0.02 (1.19)	2.03 (0.55) *
Private clinics/hospital	1.40 (0.42) *	1.238 (0.547)	1.42 (0.49)	1.16 (.70)	1.52 (0.68)	0.92 (0.43)	0.51 (0.54)	1.55 (0.64)	−0.08 (1.05)	1.27 (0.54)
Other	1.60 (0.51)	1.09 (0.646)	2.59 (0.81)	−0.83 (0.47)	1,86 (0.78)	0.57 (0.49)	0.17 (0.63)	1.21 (0.69)	−2.20 (1.74)	1.14 (0.66)
I don’t use any health facility	1.76 (2.53)	−0.962 (1.06)	6.09 (6.50)	NA	NA	−0.09 (1.39)	−1.95 (0.51)	2.98 (3.51)	NA	NA
**Who is the most important source of information about health issues?**										
FCHVs (Ref)	(M = 10.14)	(M = 10.41)	(M = 9.57)	(M = 9.51)	(M = 9.63)	(M = 8.97)	(M = 9.30)	(M = 8.26)	(M = 8.65)	(M = 8.88)
	MD (SE)	MD (SE)	MD (SE)	MD (SE)	MD (SE)	MD (SE)	MD (SE)	MD (SE)	MD (SE)	MD (SE)
Doctor	0.68 (0.40)	0.73 (0.41)	0.26 (0.58)	0.65 (0.75)	1.06 (0.95)	0.97 (0.38)	0.87 (0.48)	0.96 (0.62)	1.44 (0.44) *	1.27 (0.50)
Health Assistant	2.50 (2.42)	3.09 (3.42)	2.04 (2.69)	NA	NA	2.22 (1.80)	2.87 (3.04)	1.74 (1.79)	NA	NA
Health Worker	0.97 (0.47)	0.60 (0.51)	1.85 (0.98)	2.81 (1.56)	2.13 (1.15)	0.94 (0.39)	0.68 (0.48)	1.48 (0.69)	1.31 (0.46) *	0.98 (0.56)
Nurse/Midwife	0.73 (0.85)	0.53 (0.81)	1.10 (2.50)	−0.96 (0.64)	−1.54 (0.78)	0.60 (0.68)	0.70 (0.78)	0.07 (1.37)	1.49 (1.01)	1.18 (0.97)
Traditional Healers	−0.06 (0.56)	−0.04 (0.70)	−0.28 (0.85)	NA	NA	0.34 (0.63)	0.60 (0.81)	−0.55 (0.65)	NA	NA
Medical shop	2.05 (0.60) *	2.01 (0.73)	1.94 (1.06)	NA	NA	2.22 (0.53) *	1.75 (0.62)	3.36 (1.12)	NA	NA

M = Mean for the reference category; MD = Mean difference from reference category; * *p* < 0.05; ** *p* < 0.01; *** *p* < 0.001.

**Table 6 ijerph-16-01369-t006:** Tobit regression results of approachability healthcare providers and healthcare delivery by depression and anxiety.

Healthcare System Quality and Approachability	Anxiety									
	Total		Male		Female		Income < 20,000 NPR		Income ≥ 20,000 NPR	
	B	SE B	B	SE B	B	SE B	B	SE B	B	SE B
**The delivery system of health care in your community is**										
Intercept	7.97 ***	0.74	7.82 ***	0.91	7.98 ***	1.30	7.40 ***	0.92	8.98 ***	2.28
Quality of healthcare in community	−0.96 ***	0.25	−1.08 ***	0.31	−0.81 +	0.43	−1.15 ***	0.31	0.39	0.72
Log (scale)	1.35 ***	0.03	1.33 ***	0.03	1.38 ***	0.05	1.35 ***	0.03	1.28 ***	0.07
Number of Observations	724		514		210		453		114	
**How approachable are the health service providers?**										
Intercept	12.05 ***	0.76	11.91 ***	0.94	12.00 ***	1.31	12.05 ***	0.99	9.94 ***	1.79
Approachability of healthcare in community	−0.40 +	0.23	−0.29	0.29	−0.52	0.39	−0.40	0.31	0.08	0.53
Log (scale)	1.37 ***	0.03	1.35 ***	0.03	1.39 ***	0.05	1.38 ***	0.04	1.28 ***	0.07
Number of Observations	750		532		218		475		115	
	**Depression**									
	**Total**		**Male**		**Female**		**Income < 20,000 NPR**		**Income ≥ 20,000 NPR**	
	**B**	**SE B**	**B**	**SE B**	**B**	**SE B**	**B**	**SE B**	**B**	**SE B**
**The delivery system of health care in your community is**										
Intercept	11.15 ***	0.74	11.61 ***	0.93	10.35 ***	1.18	11.85 ***	0.95	10.41 ***	2.33
Quality of healthcare in community	−0.48 *	0.23	−0.56 +	0.30	−0.41	0.38	−0.68 *	0.30	−0.29	0.74
Log (scale)	1.28 ***	0.03	1.28 ***	0.03	1.25 ***	0.04	1.32 ***	0.03	1.30 ***	0.07
Number of Observations	724		514		210		453		114	
**How approachable are the health service providers?**										
Intercept	11.14 ***	0.79	10.55 ***	0.89	11.78 ***	1.11	11.32 ***	0.93	9.16 ***	1.81
Approachability of healthcare in community	−0.47 *	0.21	−0.21	0.28	−0.85 *	0.34	0.51 +	0.29	0.10	0.55
Log (scale)	1.28 ***	0.03	1.29	0.03	1.23 ***	0.05	1.32 ***	0.03	1.30 ***	0.07
Number of Observations	750		532		218		475		115	

Note: + *p*-values < 0.10. * *p*-value < 0.05. ** *p*-value < 0.01. *** *p*-value < 0.001.

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
