# Peer review of "Investigating the Aftershock of a Disaster: A Study of Health Service Utilization and Mental Health Symptoms in Post-Earthquake Nepal"

_ijerph, 2019, doi:10.3390/ijerph16081369_

Round 1
Reviewer 1 Report
This paper is meaningful and quite well written, especially the Introduction. However, some points in the Methods and Results need clarification and elaboration.
1. In the Methods section, it is unclear how alternate households in the transect walk of the first household’s 0.5 km radius was selected until reaching 50. More descriptions are needed. Were they randomly selected within the 0.5kn radius? Or recruiting from those closest to the first household?
2. Was DASS translated to local languages by Social Science Baha in Nepal? What were the languages? Were the questionnaires mainly completed by the respondents themselves or with the help of the interviewers?
3. Suggest to standardize the percentages to be in 1 decimal place.
4. 97% of interviewees reported their house was damaged or destroyed. The intensity between the two conditions could be quite different. Any separate percentages for these?
5. In the Results section, are district hospitals considered as primary or secondary care settings in Nepal? How are they compared with private clinics/hospitals in term of size, manpower and quality? Some descriptions of them, as well as female community health volunteers (FCHVs), can enhance international readers’ understanding of the context in Nepal. Furthermore, are there psychiatrists, psychologists, mental health nurses, social workers and other counsellors in the district hospitals? Is mental health support mainly by GPs, doctors in other specialists, general nurses and volunteers there? It is unclear what type of health professionals the distressed consult in hospitals.
6. One minor point. The colors of the bars in figure 1 are a bit confusing as the same colors are used for different items in the two questions, and vice versa.
7. In Table 4, some items under income columns are marked with NA. Are the subsamples too small for analysis?
8. In the Discussion section, the authors may compare the depression and anxiety scores after about a year of earthquake with relevant studies in Asia and Western countries. This can indicate whether the effect of earthquake on mood lasted longer in Nepal, and reflect the level of mental health support. Besides, are there cut-off scores to decide the percentage of respondents with clinical/sub-clinical depression or anxiety? The authors may consider listing the percentages of respondents with different levels of depression and anxiety according to DASS in Table 1.
Author Response
Reviewer 1 | Response |
1. In the Methods section, it is unclear how alternate households in the transect walk of the first household’s 0.5 km radius was selected until reaching 50. More descriptions are needed. Were they randomly selected within the 0.5kn radius? Or recruiting from those closest to the first household? | In each VDC, 50 households were selected using systematic random sampling. This was done by selecting the first household closest to the city center and then every alternate household in the 0.5 km radius until 50 individuals in each VDC were included |
2. Was DASS translated to local languages by Social Science Baha in Nepal? What were the languages? Were the questionnaires mainly completed by the respondents themselves or with the help of the interviewers? | Yes it was translated into Nepali: Please see page 6. On page 5 see: All surveys were translated to Nepali by SSB and surveys were read by the trained researchers to the respondents to ensure comprehension. |
3.Suggest to standardize the percentages to be in 1 decimal place. | This has been completed |
4. 97% of interviewees reported their house was damaged or destroyed. The intensity between the two conditions could be quite different. Any separate percentages for these? | This should actually be home destroyed. This has been corrected on page 8 |
5. In the Results section, are district hospitals considered as primary or secondary care settings in Nepal? How are they compared with private clinics/hospitals in term of size, manpower and quality? Some descriptions of them, as well as female community health volunteers (FCHVs), can enhance international readers’ understanding of the context in Nepal. Furthermore, are there psychiatrists, psychologists, mental health nurses, social workers and other counsellors in the district hospitals? Is mental health support mainly by GPs, doctors in other specialists, general nurses and volunteers there? It is unclear what type of health professionals the distressed consult in hospitals. | This has been put earlier in the document on page three and four to lay the framework for the results section. |
6. One minor point. The colors of the bars in figure 1 are a bit confusing as the same colors are used for different items in the two questions, and vice versa. | We have omitted the figure to alleviate confusion. |
7. In Table 4, some items under income columns are marked with NA. Are the subsamples too small for analysis? | Yes, the subsamples are too small |

Reviewer 2 Report
Investigating the aftershock of a disaster: A study of health service utilization and mental health symptoms in post-earthquake Nepal
This manuscript reports the importance of health care access and utilization of mental health service after the disaster in Nepal. The study methods and the process are reasonable and appropriate. The ‘Results’ and ‘Discussion’ section were sound and well written. While I think the ‘Background’ section needs further description. It is often hard to describe cultural issues concerning health behavior, which needs more detailed information for readers from different cultural context to understand the issue. Therefore, in this manuscript, it is important to provide adequate information regarding ‘Mental health services and stigma in Nepal’ and ‘Mental Health Post-Disaster Nepal’. Please note below comments:
1) In page 2, the authors described traditional healers, private pharmacies, and medical shops in Nepal. I understand that traditional healers and medical shops would be trained as healthcare providers. But I assume the private pharmacies may have professional training and governmental certification or registration for their healthcare activities. How are the roles of the private pharmacies and medical shops different?
2) The last sentence of page 2 illustrates mental health post-disaster Nepal. Please describe more detailed information regarding the coordinated effort to address the mental health of survivors. Who was assigned as the control tower, and what kind of psychosocial support sub-clusters were participated? What role did the mobile health clinics play? What school and community-based interventions, and public awareness initiatives were provided?
3) In page 4, the researchers developed healthcare access questions to identify types of available health care, quality of services, and utilization. Please describe more details of the healthcare access questions. Were they Likert scales? If so, please provide the choice of options for each question.
3) Review the manuscript as for typographer.
Author Response
Reviewer 2 | |
1) In page 2, the authors described traditional healers, private pharmacies, and medical shops in Nepal. I understand that traditional healers and medical shops would be trained as healthcare providers. But I assume the private pharmacies may have professional training and governmental certification or registration for their healthcare activities. How are the roles of the private pharmacies and medical shops different? | In rural setting of Nepal (where this study was done) there is not much difference in the private pharmacies and medical shops. Usually they are run by the paramedics (in context of Nepal paramedics are professionals who have received 1 to 3 years of medical studies, this includes Health Assistants, Auxiliary Health Workers) who work in the government health institutions ( as their business and using their skill/knowledge) as in the health posts there are limited supply of medicine, most times medicines are not available. So the paramedics running these shops do have professional knowledge and certification to prescribe medicines. In case of medicine shops selling natural herbs (ayurvedic medicines) they don’t have such criteria. Now Nepal Government is trying to regulate the paharmacies so it is compulsory for a person ta have diploma in pharmacy (3 years) or a degree in pharmacy ( 4 years). This has been described further in the manuscript. |
2) The last sentence of page 2 illustrates mental health post-disaster Nepal. Please describe more detailed information regarding the coordinated effort to address the mental health of survivors. Who was assigned as the control tower, and what kind of psychosocial support subclusters were participated? What role did the mobile health clinics play? What school and community-based interventions, and public awareness initiatives were provided? | This has been added on page 4: Mobile health clinics, for example, offered temporary counseling and outreach to remote earthquake affected regions. School and community-based interventions focused on empowering parents to become involved in school activities, and training teachers in positive discipline approaches to create a respectful and safe learning environment. Awareness initiatives involved raising public awareness on mental health issues and reducing stigma (19). |
3) In page 4, the researchers developed healthcare access questions to identify types of available health care, quality of services, and utilization. Please describe more details of the healthcare access questions. Were they Likert scales? If so, please provide the choice of options for each question. | We have added the response set for each question. Please note two questions were categorical, two were Likert, and one was dichotomous (yes/no) |
3) Review the manuscript as for typographer. | This has been done |
